



**Drought-induced biomass burning as a source of black carbon to the Central**
**Himalaya since 1781 CE as reconstructed from the Dasuopu Ice Core.**
Joel D. Barker[1,2], Susan Kaspari[3], Paolo Gabrielli[1,2], Anna Wegner[2], Emilie Beaudon[2],
M. Roxana Sierra-Hernández[2], Lonnie Thompson[1,2]
[1] Byrd Polar and Climate Research Center, The Ohio State University, Columbus,
43210, USA.
[2] School of Earth Sciences, The Ohio State University, Columbus, 43210, USA
[3] Department of Geological Sciences, Central Washington University, Ellensburg, USA,

11 98926

*Correspondence to: Joel D. Barker (barke269@umn.edu)*
**Abstract**
Himalayan glaciers are melting due to atmospheric warming with the potential to limit
access to water for more than 25% of the global population that reside in these glacier
meltwater catchments. Black carbon has been implicated as a factor that is contributing
to Himalayan glacier melt, but its sources and mechanisms of delivery to the Himalayas
remain controversial. Here, we provide a 211-year ice core record spanning 1781 –
1992 CE for refractory black carbon (rBC) deposition from the Dasuopu glacier ice core,
that has to date provided the highest elevation ice core record (7200 m). We report an
average rBC concentration of 1.5 µg/L (SD = 5.0, n = 1628) over the 211-year period.
An increase in the frequency and magnitude of rBC deposition occurs after 1877 CE,
accompanied by decreased snow accumulation associated with a shift in the North
Atlantic Oscillation Index to a positive phase. Typically, rBC is deposited onto Dasuopu
glacier during the non-monsoon season, and short-lived increases in rBC concentration
are associated with periods of drought within neighboring regions in north-west India,
Afghanistan and Pakistan. Using a combination of spectral and back trajectory
analyses, and comparison with a concurrent analysis of trace metals at equivalent
depths in the same ice core, we show that biomass burning resulting from dry



conditions is a source of rBC to the central Himalaya, and is responsible for deposition
that is up to 60 times higher than the average rBC concentration over the time period
analyzed. We suggest that biomass burning is a significant source of rBC to the central
Himalaya, and that the rBC record can be used to identify periods of drought in nearby
regions that are up-wind of Dasuopu glacier.
**1 Introduction**
Although the rate and extent of glacier melt differs geographically, the overall trend of
glacier mass loss globally, and particularly in mountain glaciers, is well documented
(IPCC, 2013). While warming summer temperatures resulting in increased glacier mass
loss and decreasing precipitation as snow are important factors contributing to glacier
mass wastage (Sakai and Fujita, 2017), the deposition of atmospheric aerosols that
darken the glacier surface also contribute to melt (Flanner et al., 2007; Xu et al., 2009).
The most efficient of these aerosols is black carbon (BC) which is produced by a variety
of combustion processes (Bond et al., 2004, 2013), most commonly by the incomplete
combustion of fossil fuels and biomass (Jacobson, 2004; Hammes et al., 2007). BC is
also the dominant absorber of visible light in the atmosphere (Lindberg et al., 1999) and
exerts a positive radiative forcing globally, second only to $CO_2$ (+1.1 W $m^{-2}$ and +1.6 W
$m^{-2}$ respectively; Ramanathan and Carmichael, 2008). BC continues to absorb radiation
upon deposition from the atmosphere onto glacier surfaces, reducing ice and snow
albedo, leading to melt (Hansen and Nazarenko, 2004; Forster and Ramaswamy, 2007;
Xu et al., 2009; Doherty et al., 2013).

A significant source of BC emitted to the atmosphere results from anthropogenic activity
(Ramanathan and Carmichael, 2008; Bond et al., 2013). The BC flux to the atmosphere
has increased by a factor of 2.5 since the European Industrial Revolution, resulting in an
increase of the global atmospheric BC burden by a factor of 2.5 - 3 (Lee et al., 2013).
BC's relatively short atmospheric residence time influences its distribution globally, with
the highest concentrations being proximal to BC emission sources (Bond et al., 2007;
Xu et al., 2009; Bond et al., 2013). Asian regions surrounding the Himalaya are major
sources of atmospheric BC (Novakov et al., 2003; Bond et al., 2007; Ramanathan et al.,





2007; Bond et al., 2013) and southern Himalayan glaciers are particularly influenced by
BC emissions from India (Kopacz et al., 2011; Gertler et al., 2016) and more local
emission sources that may add to the broader-scale regional flux (Kaspari et al., 2011).

Atmospheric aerosols, including BC, are warming the cryosphere and accelerating snow
melt in the western Tibetan Plateau and Himalayas (Lau et al., 2010) and altering the
regional hydrologic cycle (Immerzeel et al., 2010). This is a concern because Hindu
Kush Himalayan (HKH) glacier melt affects the water security, particularly during the
early- and post-monsoon season (Hill et al., 2020), of densely populated regions of
south-east Asia. Meltwater from HKH glaciers are the source of ten major rivers that
provide water for irrigation, hydropower, and ecosystem services for two billion people
across Asia (Scott et al., 2019); over 25% of the global population.

Research into BC's interaction with the HKH cryosphere has increased in recent years.
Several studies have documented the magnitude and timing of BC deposition using
short-term BC records preserved in surface snow that span 1 - 2 years (e.g. Xu et al.,
2009; Ming et al., 2008; Ming et al., 2012; Kaspari et al., 2014; Zhang et al., 2018; Thind
et al., 2019). More recently, continuous surface measurements of near-surface
aerosols, including BC, have been reported for the HKH region (e.g. Marinoni et al.,
2010; Bonasoni et al., 2010; Cao et al., 2010; Babu et al., 2011; Chaubey et al., 2011;
Marinoni et al., 2013; Niu et al., 2017; Negi et al., 2019). While useful for tracking the
evolution of atmospheric BC at high temporal resolution, these studies do not provide a
longer-term historical context against which current levels of BC can be compared.
Records of BC deposition preserved in ice cores are useful as longer-term
environmental archives for reconstructing atmospheric aerosol composition that span
decades (Liu et al., 2008; Ming et al., 2008; Ginot et al., 2014). In the HKH region, these
archives are essential for identifying trends in BC deposition onto HKH glaciers in
response to increasing BC emissions in surrounding regions. For example, Ming et al.
(2008) report an increasing trend in BC deposition onto East Rongbuk Glacier (Mt.
Everest; 6500 m above sea level (asl)) during a 10 year period beginning in 1965, and
then another increase beginning in 1995 to the end of the record in 2001. Xu et al.



(2009) report a period of relatively high concentrations in the 1950s and 1960s in 4
Himalayan and Tibetan Plateau glaciers (Muztagh Ata, Guoqu glacier, Noijin Kangsang
glacier, East Rongbuk glacier, and Tanggula glacier) and suggest a European source of
BC to these sites. They also note an increase in BC on the eastern-most site (Zuoqiupu
glacier) beginning in the 1990s and suggest an Indian source of BC for the region.
Similarly, Liu et al. (2008) report high elemental carbon (a form of BC) at Muztagh Ata
from 1955 - 1965. Ginot et al. (2014) report BC concentrations at Mera Glacier from
1999 - 2010, and suggest that variations in BC over this period respond primarily to
monsoonal rather than anthropogenic forcing.

Kaspari et al. (2011) were the first to present a BC record that extended back to the pre-
industrial period (1860-2000) in an ice core from East Rongbuk glacier (6518 m asl) and
reported a threefold increase in BC deposition since 1975, indicating that anthropogenic
BC is contributing to the BC flux to the southern Himalaya. Jenkins et al. (2016) report
an increase in BC deposition in the central Tibetan Plateau beginning in 1975 from the
Guoqu glacier ice core record spanning 1843-1982. These deep ice core records are
valuable for evaluating long-term trends in BC spanning the Industrial Revolution to the
present and the concomitant increase in anthropogenically-sourced BC emissions.
Additional ice core-derived BC records that span the period of industrialization in Asia
are required to both corroborate existing historical records of BC deposition onto HKH
glaciers and to establish a regional baseline record for BC fluxes onto the region. These
records are currently lacking for the HKH and are essential for identifying regional-scale
trends in BC deposition.

The highest elevation ice core record ever obtained is the Dasuopu ice core (C3;
Thompson et al., 2000), which was retrieved from the Dasuopu Glacier in the Central
Himalaya (28.38 °N, 85.72 °E; Fig. 1) in 1997 at an elevation of 7200 m asl. Thompson
et al. (2000) determined that monsoonal precipitation is responsible for the net
accumulation of snow onto the glacier surface, in the order of 1000 mm water
equivalent per year (in 1996), permitting an annually resolved environmental record
spanning 1440 - 1997 CE (Thompson et al., 2000). The remote location and high



elevation of the Dasuopu ice core drill site suggests that any local influence on the
deposition of atmospheric aerosols onto the glacier surface is minimal, and that
accumulation is representative of mixed free tropospheric composition (Kumar et al.,
2015). Evidence suggesting that the Dasuopu glacier differs from lower-elevation
glaciers in the region with respect to seasonal meteorology supports the hypothesis that
the flux of aerosols onto the glacier surface may be more representative of free
tropospheric composition rather than being affected by local (valley-scale)
meteorological conditions (Li et al, 2011).

Here, we quantify refractory BC (rBC; a subset of the broader BC descriptor of
carbonaceous particles that it is measured by laser induced incandescence specifically;
Lack et al., 2014) in a section of the Dasuopu ice core from 1781-1992 CE at annual to
sub-annual resolution in the glacier ice portion. We employ spectral analysis of the rBC
ice core time series to identify trends in rBC deposited onto Dasuopu glacier across
several temporal scales and to avoid "peak-picking" that might lead to subjectively
identifying episodes of increased rBC in the ice core time series. The rBC record is
compared to trace-element analysis of samples from equivalent depths along the same
ice core, as described by Gabrielli et al. (2020), and an atmospheric back trajectory
analysis to elucidate the broader-scale trends of deposition and potential rBC sources to
the southern Himalaya.

**2 Methods**
**2.1 The Dasuopu Ice Core**
Dasuopu glacier descends to the north from Mt. Xixiabangma in the Central Himalaya
(Fig. 1). The ice core examined here was drilled from the Dasuopu glacier surface to
bedrock (145.4 m) with an electromechanical drill, without using drilling fluid, and
provides a continuous record of deposition onto the glacier surface from 1010 to 1997
CE (Thompson et al. 2000). Here, we examine the upper section of the C3 ice core
(hereafter referred to as the "Dasuopu core"), from 8.4 - 120.3 m depth from the
surface, corresponding to the period 1781 - 1992 CE. Sections of the Dasuopu core
outside of this interval were not available for analysis. We use the Thompson et al.


(2000) chronology that was established using $\delta^{18}O$, dust, and $NO_3^-$ measurements, as
well as annual layer counting confirmation using the location of the 1963 CE beta
radioactivity peak from thermonuclear tests at a depth of 42.2 m to determine the core's
age-depth relationship. Thompson et al. (2000) also used two major monsoon failures
(1790-1796 and 1876-1877) as age/depth benchmarks that are reflected in the dust and
Cl$^-$ records to validate the ice core dating chronology. The chronology is accurate to $\pm 3$
years (Thompson et al., 2000).

**2.2 Sample Preparation**
A portion of the Dasuopu core has been housed in the Ice Core Storage Facility (Byrd
Polar and Climate Research Center (BPCRC)) at -30 °C since the original analysis by
Thompson et al. (2000). The portion of the Dasuopu core analyzed here is
characterized by consolidated firn from 8.4 m - 56.4 m and glacier ice from 56.4 - 120.3
m depth. Ice was sampled continuously (with the exception of intervals noted in Suppl.
Info. Table 1) in a cold room (-5 °C) at sub-annual resolution (2.5 - 10 cm sample
interval) with a band saw along the length of the ice section. Each ice sample was
divided in half to permit the analysis of BC and trace elements from identical depths
throughout the core (n = 1572). Prior to rBC analysis, each ice sample was rinsed with
MQ water at room temperature in a class 100 laboratory to remove any contaminants
from the outer edges of the core, placed in a sealable polyethylene bag and
immediately stored frozen (-34 °C) to ensure that the sample did not melt prior to
analysis.

Due to sample volume limitations resulting from previous studies of the Dasuopu core
(e.g. Thompson et al., 2000; Davis et al., 2005), 56 firn samples (5.5 - 10 cm length)
were collected at discontinuous intervals (where sufficient sample volume was
available) from 8.4 - 56.4 m depth in the cold room (-5 °C) using a band saw. The outer
2 cm of each sample (n = 56) was removed using clean stainless-steel knives (soaked
in $HNO_3^-$ and rinsed with MQ water) under laminar flow conditions in the cold room to
remove surface contaminants. Clean firn samples were stored frozen (-30 °C) in double
Ziploc bags until analysis.






### 2.3 BC analysis

**2.3 BC analysis**
rBC was quantified by laser induced incandescence using a Single Particle Soot
Photometer (SP2; Droplet Measurement Technologies, Longmont, U.S.A.; Schwarz et
al., 2006; Wendl et al., 2014) at Central Washington University (Ellensburg, WA, USA).
Frozen samples were melted at room temperature, transferred from storage bags into
50 ml polypropylene centrifuge tubes, and sonicated for 20 minutes immediately prior to
analysis. Each liquid sample was stirred with a magnetic bar as water was routed into a
CETAC U-5000AT+ ultrasonic nebulizer (Teledyne CETAC Technologies, Omaha,
U.S.A.) using a peristaltic pump. The resultant aerosols flowed to the SP2 inlet at a
known rate using carbon-free air carrier gas. The peak intensity of light emitted by an
incandescing rBC particle is linearly proportional to its mass (Schwarz et al., 2006), and
the SP2 detects this emitted light using the amplified output from 2 photodetectors
(broadband and narrowband) to provide a detection range of ~70 – 500 nm volume-
equivalent-diameter (VED; Kaspari et al., 2014). A 5-point calibration curve using
Aquadag standards and MQ water was performed daily to correct for BC loss during
nebulization (Wendl et al., 2014). MQ water was analyzed every 5 samples as a blank
to monitor instrument baseline conditions. If the baseline was above background levels,
then MQ water would be run through the system until stability was achieved. Baseline
instability was not observed throughout the course of the analysis. The SP2 data output
was processed using the PSI SP2 Toolkit ver. 4.100a (Paul Scherrer Institut, CH) and
the IGOR Pro software platform (WaveMetrics Inc., Portland, U.S.A.).

### 2.4 Spectral analysis

**2.4 Spectral analysis**
The record of rBC concentration with depth through the Dasuopu ice core provides a
time series of rBC deposition onto Dasuopu glacier over time. The decomposition of the
time series into time-frequency space using spectral analysis (wavelet analysis) permits
the identification of dominant modes of variability and their variance with time (Torrence
and Compo, 1998). Wavelet analysis is well suited to the analysis of time series data
where frequency and/or magnitude is non-stationary through the signal (Debret et al.,
2007). For example, wavelet analyses have been used to identify Himalayan climatic



oscillations related to orbital forcing, Dansgaard-Oeschger cycles, and Heinrich events
in the 1992 Guliya ice core (Yang et al., 2006), characterize the increased role of ENSO
climate forcing on Antarctic temperature since ~1850 from ice core records from East
and West Antarctica (Rahaman et al., 2019), and a switch from external forcing to
internal forcing mechanisms on global climate during the mid-Holocene (Debret et al.,

223  2009).


The ice core sampling strategy employed here may influence the results of the spectral
analysis because the uppermost firn section was not sampled continuously and there
are occasional sampling gaps in the glacier ice section (Suppl. Info. Table 1). The
discontinuous sampling of the firn section likely resulted in an incomplete
characterization of rBC deposition onto the Dasuopu glacier since 1944 (56.4 m depth).
Further, the number of samples per year is not consistent throughout the record
because of interannual differences in snow accumulation (Suppl. Info. Fig. 1). It is
important to note that the spectral analysis treats the rBC time series as a linear depth-
time function. However, because the depth-time relationship in the ice core is not linear,
data is treated here as a function of the sample number of progression with depth in the
ice core, while the dates of the individual features detected relative to sample number
are specified using the Thompson et al. (2000) depth-age model. Therefore, the
spectral decomposition of the time series into time-frequency space is achieved while
minimizing the influence of data gaps and non-linear accumulation rate.

The wavelet analysis of the Dasuopu rBC record was performed using the Wavelet
Toolbox in Matlab (ver. R2020a; Mathworks). A continuous 1-D wavelet transform was
generated to identify modes of variability and the characteristics of that variability with
time throughout the rBC record. The Mexican Hat (or Rickler) mother wavelet was
chosen because it is similar to the shape of the annual variability in the rBC
concentration signal across the time series (Suppl. Info. Fig. 2).


**2.5 Trace element analysis**


Trace element quantification at equivalent depths as the rBC was only possible for the
glacier ice section of the Dasuopu ice core due to lack of sampling volume in the
corresponding overlying firn sections. Trace element concentration was determined by
Inductively Coupled Plasma Sector Field Mass Spectrometry (ICP-SFMS) at BPCRC.
Twenty three trace elements were measured (Al, As, Ba, Bi, Cd, Co, Cr, Cs, Fe, Ga,
Mg, Mn, Mo, Nb, Ni, Pb, Rb, Sb, Ti, Tl, U, V, and Zn) using methods described in
Uglietti et al. (2014). The trace element crustal enrichment factor (EF) is used to identify
trace element contributions exceeding natural background levels, and was calculated
relative to Fe and elemental ratios of dust from the Tibetan Plateau following Gabrielli et
al. (2020) as an additional variable to be compared with rBC.

**2.6 Back trajectory analysis**
While the complex geomorphology of the Himalayas affects local wind patterns, back
trajectory modeling permits the characterization of the broader regional catchment from
which rBC may be derived. Atmospheric circulation capable of delivering rBC to
Dasuopu glacier was identified using the Hybrid Single Particle Lagrangian Trajectory
Model (HSYPLIT; NOAA Air Resources Laboratory, 2018). A 7-day back trajectory was
chosen as a conservative estimate of rBC atmospheric residence time given the range
reported in the literature (e.g. Ogren and Charlson, 1983; Reddy and Boucher, 2004,
2007; Samset et al., 2014; Lund et al., 2018). Back trajectories from the Dasuopu drill
site were calculated at 6 hour intervals from 1948-1991 for January (winter/non-
monsoon) and July (summer/monsoon) using the NCEP/NCAR (National Centers for
Environmental Prediction/National Center for Atmospheric Research) reanalysis from
1948 (the limit of the NCEP/NCAR dataset) to 1991.

**3 Results**
**3.1 The rBC record**
Figure 2a shows the 211-year rBC record from the Dasuopu ice core. The mean rBC
concentration is 1.5 µg/L (SD = 5.0, n = 1628) from 1781 (± 3 years) to 1992 CE. The
mean rBC concentration in the glacier ice section from 1781 to 1944 and the
discontinuously sampled firn section from 1944 to 1992 is 1.4 µg/L (SD = 4.4, n = 1572)





280 and 6.0 µg/L (SD = 13.2, n = 56), respectively. Note that the median values for the

281 same time periods are less influenced by outliers with high concentrations (median

282 1781 to 1944 = 0.2 µg/L, 1944 to 1992 = 0.6 µg/L). Even though the rBC concentration

283 in the ice and firn described here is significantly different (two tailed Mann-Whitney U

284 test, $p<0.05$), the effect of discontinuously sampling the firn section and its accurate

285 characterization of rBC since 1944 is unknown. It is possible that the firn section is

286 biased towards higher rBC concentrations because only 26% (14 of 54) of the firn

287 samples correspond to snow deposited during monsoon conditions, as indicated by

288 depleted $\delta^{18}$O (data not shown), that is a period associated with lower atmospheric

289 aerosol loading (Lelieveld et al., 2018). In general, rBC deposition corresponds to $\delta^{18}$O

290 enrichment (Fig. 3) and increased dust in glacier ice, indicating that rBC is deposited

291 during the non-monsoonal dry season (Fig. 3; Kaspari et al., 2014). Occasional

292 exceptions occur, for example in 1824 CE, when a period of high rBC deposition

293 corresponds to a relatively low dust concentration and a low $\delta^{18}$O value (Fig. 3a).

294

295 The smoothed (5 year median) rBC concentration and flux (the product of mean annual

296 rBC concentration and snow accumulation) records show an increase beginning in

297 ~1870 and again in ~1940 (Fig. 2a, b). The discontinuous firn section of the core has

298 elevated concentrations during the late 1960s - 1970s, consistent with observations

299 from East Rongbuk glacier by Ming et al. (2008) and Kaspari et al. (2011), and for

300 Tanggula glacier by Xu et al. (2009).

301

302 **3.2 Spectral Analysis**

303 The spectral analysis of the rBC record identifies three modes of variability (Fig. 4b):

304   First, the mode at a = 6 (a = 0.5 x frequency) indicates high frequency, and

305 generally, relatively low amplitude variability in spectral coefficients ( 81% of rBC

306 concentrations are < 1 µg/L) occurring at ~ annual (12 data points/year; SD = 4.3, n =

307 112) resolution with isolated relatively higher amplitude events dispersed throughout the

308 record (Fig. 4c). The frequency of these higher amplitude events increases from ~1877

309 until 1992 CE (Fig. 4a, c).





Second, a lower frequency mode (a = 27; ~4.5 years) captures periodic peaks in

rBC concentrations centered at 1825, 1877, 1888, 1908, 1920, 1930, and 1977 CE if
peaks that are >25% of the largest peak's amplitude in the time series (1977 CE) are
considered (Fig. 4d). Dips in the a = 27 spectral coefficients indicating periods of low
amplitude (defined here as >25% of the amplitude of the lowest dip at 1937 CE), occur
at 1818, 1868, 1875, 1880 - 1884, 1893, 1914, 1924, and 1937 CE (Fig. 4d).

Third, the a = 512 (~85 year) mode identifies a shift from samples with negative

spectral coefficients to those with positive spectral coefficients at 1877 CE (Fig. 4e). All
three modes identify a period early in the rBC record characterized as a quiescent
period (1781 - 1877 CE) where rBC concentrations do not exceed 19.3 µg/L (mean =
0.8, SD = 3.0, n = 880), except for the isolated peak (63.3 µg/L) at 1825 CE (Fig. 4a, c,
d)). Prior to 1877 the rBC concentration in the ice core is significantly lower (Mann-
Whitney U test, $p < 0.05$) and less variable (mean = 0.8, SD = 3.03, n = 898) than the
post 1877 period (mean = 2.3, SD = 6.6, n = 732; Fig. 4). While the ~85 year mode
identifies a shift from negative to positive spectral coefficients in 1877, the 5 year
median of the rBC record identifies an increase occurring at ~1870. This suggests that
the wavelet analysis may be sensitive to individual or tightly clustered peaks in the rBC
record, such as those that occur between 1875 and 1880 (Fig. 3a, 4a).

**3.3 Comparison of the rBC record with the trace element record**
When considering the full record (n = 857 to 916 depending on the element; Table 1), all
of the trace element concentrations analyzed are significantly correlated with rBC
(range of 0.15 (Zn; n = 915) to 0.27 (Rb; n = 914); Table 1; $\alpha$ = 0.01; Spearman
correlation test is used instead of Pearson correlation test because the rBC and trace
element data are not normally distributed). If the low-rBC pre-1877 period, as indicated
by the spectral analysis, is considered independently, then the correlation between
trace element and rBC are still statistically correlated (range of 0.26 (Zn, n = 915) to
0.44 (Mg and Mn, n = 915). In contrast, the post 1877 period shows a statistically
insignificant slightly negative correlation between the trace elements and rBC ranging
from -0.04 (Cs and Nb, n = 913 and 915, respectively) to -0.10 (Bi and Mn, n = 857 and
915, respectively).



The crustal enrichment factor (EF) for all of the trace elements were significantly weakly
to moderately negatively correlated with rBC for all trace elements, ranging from -0.21
for Mg to -0.57 for Ga, except for Mn which was insignificantly positively correlated
(0.02). The trace element EFs were more negatively correlated to rBC during the post
1877 period than the pre-1877 period (excluding Mn because it was insignificantly
correlated; SD = 0.14), although this difference is not statistically significant, $t(22) =$
1.88, $p$ = 0.07 ($p$ < 0.05).

**3.4 Back trajectory**
Figure 5a shows the results of the July back trajectory showing that aerosols are
primarily derived from areas to the south-west of the Dasuopu drill site, from the Arabian
Sea and across western and northern India during the monsoon. A secondary source is
located to the west and draws atmospheric aerosols from the eastern Mediterranean
Sea and Arabian Peninsula. January (non-monsoon) circulation is derived from the
westerly circulation across north-eastern Africa, Central Europe, the Arabian Peninsula
and north-west India (Fig. 5b).

**4 Discussion**
**4.1 rBC concentrations**
The mean rBC concentration in the Dasuopu ice core, from 1781 to 1992 CE is 1.5 μg/L
(SD=5.0, n=1628); 6 times higher than the average rBC reported by Kaspari et al.
(2011) for the period 1860-1992 and ~2 times lower than BC reported by Ming et al.
(2008) and Xu et al. (2008) for the East Rongbuk ice core record over similar time
periods (Fig. 1). Note that while Kaspari et al. (2011) measured BC from the East
Rongbuk core using the same incandescence method used here, samples were stored
as liquid and measured concentrations are likely underestimated due to rBC particle
adherence onto the walls of the storage container and/or agglomeration of BC particles
above the detected particle size range (Wendl et al., 2014; Kaspari et al., 2014). In
contrast, Ming et al. (2008) and Xu et al. (2009) measured BC concentration by thermo-
optical methods, which may result in an overestimation of reported BC due to organic



matter pyrolysis during analysis (Gilardoni and Fuzzi, 2017), and a larger fraction of the
carbonaceous particles being classified as BC.

**4.2 rBC seasonality**
Seasonally, peaks in rBC concentration correspond to intervals of increased dust
concentration and enriched $\delta^{18}O$ over the entire ice core record (see examples in Fig.
3), indicating most BC deposition occurs during the non-monsoonal season when drier
westerly air masses dominate atmospheric circulation (Fig. 5). Weather station
measurements and previous snow/ice studies in the region confirm that rBC
concentrations are lower in near-surface air at the Nepal Climate Observatory-Pyramid
(NCO-P; 5079 m a.s.l.) during the monsoon (Bonasoni et al., 2010; Marinoni et al.,
2010, 2013) and higher during the pre-monsoon period (Babu et al., 2011; Nair et al.,
2013; Ginot et al., 2014; Kaspari et al., 2014; Chen et al., 2018).

**4.3 Temporal variations in rBC deposition and regional climate**
The pre-1877 CE period differs from the post-1877 CE period in the frequency and
amplitude of variability in rBC deposition (Fig. 2a and 4e). The high BC deposition event
in ~1825 CE (Fig. 4c) occurs during an otherwise quiescent pre-1877 CE period
coinciding with a time of severe regional moisture stress/droughts as reflected in
suppressed tree ring growth across Nepal, peaking in 1817 CE (Figs. 6 and 7 in Thapa
et al., 2017). This period of abnormally dry conditions occurs after 2 large volcanic
events; the Tambora eruption of 1815 (Stothers, 1984) and an eruption of unknown
origin in 1809 CE (Cole-Dai et al., 2009). Anchukaitis et al. (2010) argue that major
explosive eruptions in the tropics can disrupt the Asian monsoon system and result in
drier conditions in central Asia for up to 8 years afterward. Dry conditions are typically
associated with an increase in the frequency and severity of biomass burning in south-
east Asia (Baker and Bunyavejchewin, 2009) and the association between dry
conditions and increases in rBC deposition suggests that biomass burning may be a
source of high rBC deposition events onto Dasuopu glacier.



From ~1877 CE until the end of the rBC record in 1992, rBC concentrations are
significantly higher and the amplitude of rBC deposition increases, as indicated by the
shift from negative to positive spectral coefficients at a = 512 (Fig. 4e). This suggests a
change in either the magnitude of rBC emission source(s) or in the atmospheric
mechanism that delivers rBC to Dasuopu glacier after ~1877 CE. The post ~1877
increase in rBC corresponds to a decrease in snow accumulation onto Dasuopu glacier
(Fig. 2c; Davis et al., 2005) and an increase in the rBC flux from the atmosphere
beginning in ~1880 (Fig. 2b). This decrease in snow accumulation has been linked to a
strengthening of the Icelandic Low pressure system as temperatures in the Northern
Hemisphere warmed at the termination of the Little Ice Age (LIA). This resulted in a shift
in the North Atlantic Oscillation Index (NAO) from a negative mode to a positive mode,
contributing less moisture to the southern Himalaya during winter (Davis et al., 2005).
Less winter snow accumulation post ~1877 would be associated with drier winter (non-
monsoon) conditions generally, when rBC deposition onto Dasuopu glacier is highest.

**417  4.4 The influence of drought and biomass burning on the rBC record**

Biomass burning, and associated rBC emissions result from dry conditions and drought
that lowers the water table and dries biomass fuel (Baker and Bunyavejchewin, 2009;
Tosca et al., 2010). Further, aerosols produced during fires may contribute to a positive
feedback cycle where smoke plume shading decreases sea surface temperature, while
increased concentrations of atmospheric BC warm and stabilize the troposphere,
suppressing convection and precipitation and intensifying drought conditions on land
(Tosca et al., 2010). High BC aerosol levels in ambient air corresponding to agricultural
burning beginning in late April and forest fire activity during the non-monsoon season
was reported by Negi and others (2019) from ambient air measurements at Chirbasa,
India (Gangotri glacier valley) during 2016. The spectral coefficients calculated here
identify trends in rBC deposition onto Dasuopu glacier and can be compared to regional
rainfall data from a network of rain gauge stations that are distributed across India to
identify periods of dryness (e.g., Parthasarathy et al. 1987) associated with rBC
deposition.




Continuous regional instrumental rainfall records within the atmospheric catchment for
atmospheric aerosols to Dasuopu glacier prior to the early 1900s CE are rare, and
biomass burning records are non-existent. However, continuous tree ring-based
reconstructions of precipitation conditions for Europe, North Africa, and the Middle East
is provided by the Old World Drought Atlas (OWDA; Cook et al., 2015) and includes
areas identified by the back trajectory analysis as being potential source regions for rBC
to Dasuopu glacier (Fig. 5b). The Monsoon Asia Drought Atlas (MADA; Cook et al.,
2010) provides a similar dataset for regions in East Asia, including Pakistan and
Afghanistan, which may contribute rBC to Dasuopu glacier (Fig. 5b). An instrumental
record for both the OWDA and MADA begins in 1901 (Fig. 6). Comparing the peaks in
rBC deposition identified by the spectral coefficients (a = 27, ~4.5 year frequency)
centered at 1825, 1877, 1888, 1898, 1908, 1920, 1930, and 1977 CE (Fig. 4d) to the
reconstructed and instrumental self-calibrating Palmer Drought Severity Index (scPDSI)
for the summer season (where positive and negative scPDSI indicate wet and dry
conditions respectively; Fig. 6), it is possible to identify periods of dryness that might
contribute to the production of rBC by biomass burning.

rBC wavelet coefficient peaks in 1825 and 1877 CE occur at the end of a decade-long
period of negative scPDSI in the OWDA and MADA reconstructions, respectively (Fig.
6). Similarly, 1888, 1898, and 1930 follow years of negative scPDSI in either the OWDA
or MADA reconstructions, indicating periods of dryness preceding episodes of rBC
deposition onto Dasuopu glacier (Fig. 6). The 1908 and 1920 CE peaks do not follow
periods of negative scPDSI in the OWDA or MADA reconstructions, but follows periods
of negative scPDSI in the MADA instrumental record (Fig. 6) indicating that dryness is
associated with these rBC deposition peaks as well. The peak centered at 1977 CE
follows periods of positive scPDSI in the OWDA and MADA reconstructions and
instrumental records and does not appear to be related to abnormally dry conditions,
and may indicate an unidentified source of rBC. Conversely, dips in the spectral
coefficients at a ~4.5 year frequency (a = 27) indicate periods of low rBC deposition
occurring at 1818, 1868, 1875, 1880-1884, 1893, 1914, 1924, and 1936 CE. With the
exception of the dip centered at 1875 and 1936 CE, dips in the spectral coefficient



record follow periods of positive scPDSI in either or both the OWDA and MADA tree ring
reconstruction. While dips centered at 1914 and 1924 CE follow periods of positive
scPDSI in both the OWDA and MADA instrumental record, 1936 CE follows a period of
positive scPDSI in the MADA instrumental record only (Fig. 6).

In addition to the scPDSI from OWDA and MADA tree ring reconstructions and the
instrumental record (since 1900 CE), an independent historical record for rainfall is
available for India that was compiled by Mooley et al. (1981) and has since been
reported in terms of drought/flood severity by Parthasarathy et al. (1987; Suppl. Table
1a, b). As mentioned, several periods of high rBC concentration are identified by the
spectral coefficients at a = 27 (~4.5 year frequency) centered at 1825, 1877, 1888,
1898, 1908, 1920, 1930, and 1977 CE (Fig. 4d). These periods of high rBC deposition
coincide with periods of drought reported for India, particularly in western/northwestern
meteorological subdivisions (Parthasarathy et al., 1987) within the ±3 years dating error
of the ice core chronology (Fig. 7; Suppl. Fig. 3 a). For example, from 1876 – 1878,
India experienced widespread moderate to severe drought conditions (Parthasarathy et
al., 1987; Fig. 7a) and soil moisture deficits (Mishra et al., 2019) that resulted in the
"Madras Famine" (Cook et al., 2010; Mishra et al., 2019). In 1888 (and 1891 which is
within the ±3 year ice core dating uncertainty), regions in western and northwestern
India experienced moderate and severe drought conditions (Fig. 7b). In 1899
(corresponding to 1898 in the rBC record, ±3 years), northwest and western
meteorological subdivisions (among others) experienced severe drought while
moderate drought was experienced by most of India (Fig. 7c), resulting in famine that
affected 59.5 million people (Mishra et al., 2019). In 1911 (1908 ± 3 years) there was
extreme drought reported in the northwest and moderate drought reported in the north-
central and southwest meteorological districts (Fig. 7d). In 1918 (1920 ± 3 years), there
was severe drought reported in the north and central-west and moderate drought
reported throughout the south and north-central regions of the continent (Fig. 7e). From
1927-1929 (1930 ± 3 years), moderate drought was reported in the northern region of
India (Fig. 7f). Similar to observations from the OWDA and MADA comparisons, the
~1977 period does not stand out in the climate record as being exceptional (Suppl. Fig.



3), and it does not correspond to anomalously high rBC values (Fig. 2a) yet it
corresponds with a period of highly positive spectral coefficients (Fig. 4c, d). Finite-
length signal border effects (so called edge effects) have been well documented, where
a wavelet transform (such as that used here) yield abnormal coefficients as the wavelet
extends into the "shoulder areas" of the record that don't have data (Su et al., 2011,
Montanari et al., 2015). It is possible that the peak identified here at a = 6 and a = 27 is
a result of wavelet transform edge effects. Alternatively, sources other than biomass
burning, that have not been identified here, may contribute to high rBC values observed
in the Dasuopu ice core ~1977 CE.

Dips in the a = 27 spectral coefficient record correspond to periods of flooding in India.
For example, the trough at 1875 CE corresponds to reports of extreme flooding in the
north-west and moderate flooding in western India (Fig. 8a). It should be noted that
moderate drought was reported in the far west and south, but these conditions did not
result in a rBC peak in the a = 27 coefficients (Fig. 4d). For the period ~1880 to 1886
CE, severe and moderate flooding is reported in the west in 1884 CE, with moderate
drought to the south and east that did not result in an rBC peak in the a = 27 coefficients
(Fig. 8b). From 1880 – 1882 CE, the continent experienced relatively stable conditions
with moderate flooding in some western and north-western districts (Suppl. Info. Fig. 3).
Western India experienced severe and moderate flooding in the west and northwest in
1893 (Fig. 8c), corresponding to a dip in the a = 27 coefficients (Fig. 4d). 1914 and 1917
(1914 ± 3 years), 1926 (1924 ± 3 years) and 1933 (1936 ± 3 years) all saw severe
and/or moderate flooding in western meteorological districts, with no drought conditions
reported in the rest of India, corresponding to dips in the a = 27 coefficients (Fig. 8d, e, f
respectively).

**4.5 rBC and trace metals**
Recent work by Gabrielli and others (2020) suggested that atmospheric trace metals
preserved in the Dasuopu ice core, likely linked to the long-range transport of fine fly
ash, were indicative of emissions from coal combustion and fires used to clear forested
areas to the west of the Himalayas since the beginning of the Industrial Revolution





(~1780 CE). Fly ash is composed of alumino-silicate and iron-rich byproducts of coal
combustion and biomass burning and is enriched in trace metals (Ross et al., 2002). Fly
ash is not detected by the SP2 as configured here.

We observe a general negative correlation between BC and the crustal enrichment
factor (EF; indicative of element concentrations above the natural background derived
from crustal material) of trace metals in the Dasuopu core, particularly after 1877 CE
(Table 1) when rBC spectral coefficients are positive at a = 512 (Fig. 4e). This illustrates
that the deposition of the non-crustal fraction of trace metals (as indicated by a positive
EF), and fly ash, occurred out of phase from rBC.

rBC deposition resulting from biomass burning may be expected to correlate with trace
elements associated with the biomass source material (K, Cl, Zn, and Br; Echalar et al.,
1995). Of these, only Zn was analyzed here. Zn concentration is only weakly correlated
with rBC (0.15), although more strongly (0.26) in the pre-1877 period than in the post-
1877 period (-0.06), and Zn's EF is moderately negatively correlated, particularly in the
post 1877 period (-0.63). While the lack of correlation between potential biomass
burning-derived trace elements such as Zn and rBC might suggest a non-biomass
burning source for rBC, one should be cautious in attributing specific trace elements to
biomass burning events. For example, trace elements emitted during partial combustion
can vary depending on fire intensity (flaming vs. smoldering), fuel source (savanna vs.
forest) (Echalar et al., 1995), and size-dependent particle adhesion (Samsonov et al.,
2012). Further, biomass burning remobilizes soil-derived particles which would lower
the individual trace element's EF (Gaudichet et al., 1995) causing a negative correlation
between rBC and EFc. There is a statistically significant negative correlation with rBC to
all of the trace EFs (except for Mn), suggesting that rBC deposition is not associated
with non-crustal trace element deposition, interpreted as an indicator of fly ash
deposition (Gabrielli et al., 2020), that is enriched above the natural dust input. Of
importance is that the discontinuous sampling of firn in the Dasuopu ice core record
presented here does not capture a continuous record of rBC deposition during the post



1970s; a period when rBC is reported to have increased in the southern Himalaya
(Kaspari et al., 2011) and Tibetan Plateau (Jenkins et al., 2016; Wang et al., 2015).

**5 Conclusions**
Here, we present the highest elevation (7200 m asl) record of rBC deposition ever
reported. This record is unique in its high elevation and represents conditions in the free
troposphere, away from local sources of BC. The Dasuopu record also contributes to
the limited number of proxy records of BC deposition in the HKH region where glacier
melt, and therefore factors such as BC that affect glacier melt, influence the water
security of one of the most densely populated regions of the planet. While the Dasuopu
rBC record presented here is not well resolved during the period after the 1970s, the
record does indicate elevated BC during 1970-1980, consistent with the Everest ice
core BC record that showed elevated BC post 1970 (Kaspari et al., 2011).

rBC deposition at the Dasuopu site is highest during the winter (non-monsoon) season
when westerly circulation is dominant. Back trajectory analyses indicate that this
westerly circulation predominantly includes areas of west/northwest India, Afghanistan,
Pakistan, northern Africa, central Europe and the Mediterranean. Dry conditions
increase the production of rBC through biomass burning and we suggest that regional
biomass burning contributes to periods of high rBC deposition onto the Dasuopu glacier
during periods of dryness as indicated by historical records of precipitation within the
atmospheric catchment of Dasuopu glacier. The continuous historical record of
precipitation for India, in particular, suggests an association between moderate to
severe drought conditions in west/north-west India and rBC concentration in the
Dasuopu ice core. Upwind industrial sources of rBC, such as coal combustion, appear
to be of minor influence during these periods of increased rBC deposition as indicated
by the absence of correlation between rBC concentration in the Dasuopu core and the
crustal enrichment of industrially-sourced trace elements at equivalent depths in the ice
core. It should be noted that the Dasuopu ice core rBC record is discontinuous during
the period of increased regional industrial activity thus the available data cannot
address the importance of this regional industrialization to rBC deposition onto Dasuopu



glacier. Together, evidence presented here indicates that while rBC transport in the free
troposphere is influenced by large scale synoptic circulation, regional sources of rBC
strongly influence rBC deposition onto Dasuopu glacier, particularly after ~1877, and
that the rBC record from Dasuopu glacier may provide a proxy record for drought and
resultant biomass burning within its catchment of atmospheric circulation.

**Acknowledgments**
This work was funded by the NSF Atmospheric Chemistry Program (award # 1149239)
and the by the NSF-ESH program, The Ohio State University, the Ohio State
Committee of Science and Technology, and the National Natural Science Foundation of
China. We thank the many scientists, engineers, technicians, and graduate students
from the Byrd Polar and Climate Research Center and the Lanzhou Institute of
Glaciology and Geocryology (China) that contributed to the collection and previous
analysis of the Dasuopu ice core. We are grateful to Julien Nicolas for performing the
graphic display of the back trajectories. This is BPCRC contribution no. xxxx.

**Data availability**
The data presented in this work are archived at the National Oceanic and Atmospheric
Administration World Data Center-A for Paleoclimatology at xxxx.

**Author contribution**
Barker performed the sample preparation, BC analysis and interpretation, and was the
primary author of the manuscript. Kaspari assisted with the BC analysis and
interpretation of the BC record. Gabrielli designed the overall project, performed the
trace element analysis with Wegner. Wegner, Beaudon, and Sierra-Hernández cut the
samples from the ice core and performed the trace element analysis. Thompson
retrieved the Dasuopu ice core. All authors contributed to manuscript preparation.

**Competing interests**
The authors declare that they have no conflict of interest.





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





Figure 1:

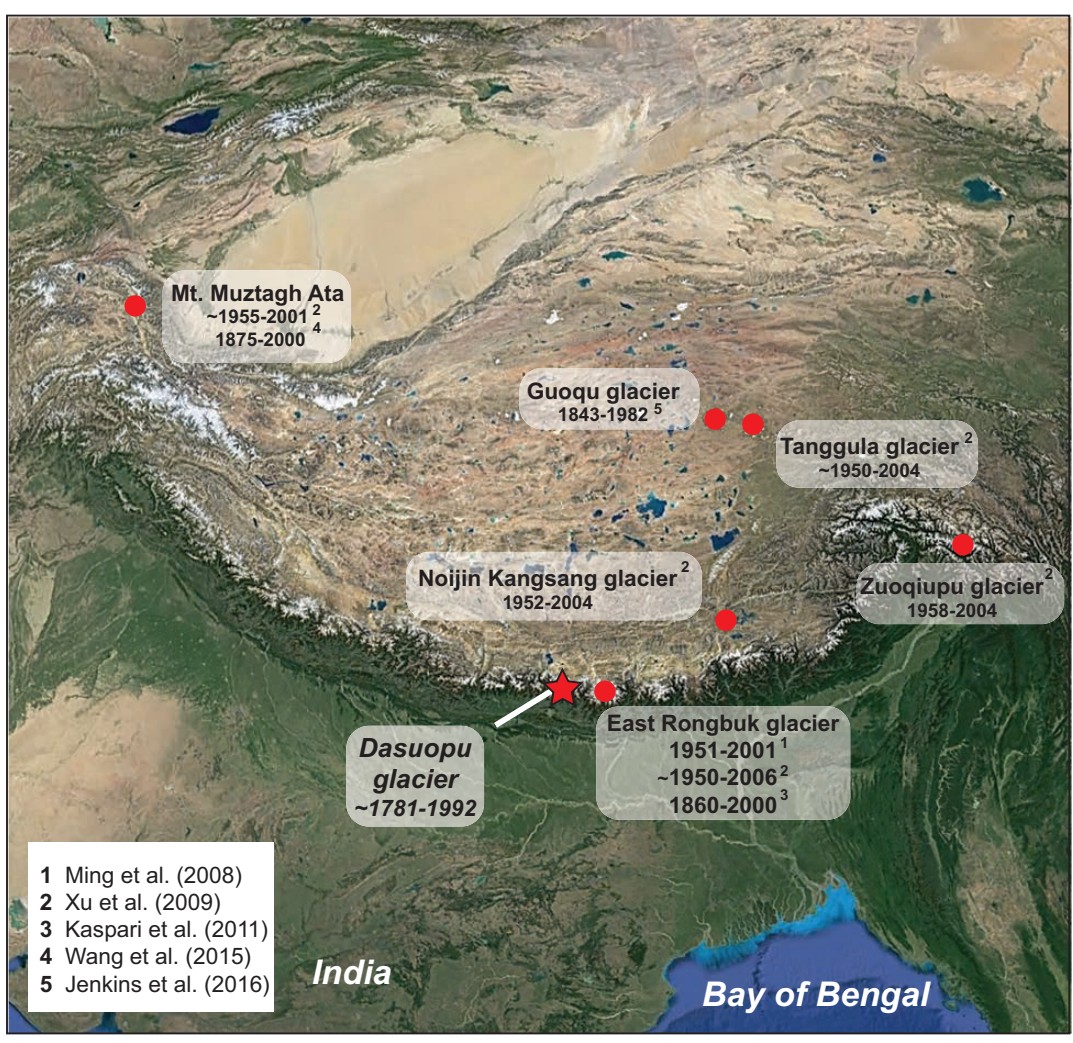

Source: "Tibetan Plateau"  28.38°N, 85.72°E.  © **Google Earth**, Image: Landsat / Copernicus. 11/30/2016. 11/20/2019.
Fig. 1: The location of Dasuopu glacier, Mt. Xixiabangma and the location of other ice
cores that have provided a historical record of BC deposition in the region. The span of
each BC record is indicated.



Figure 2:

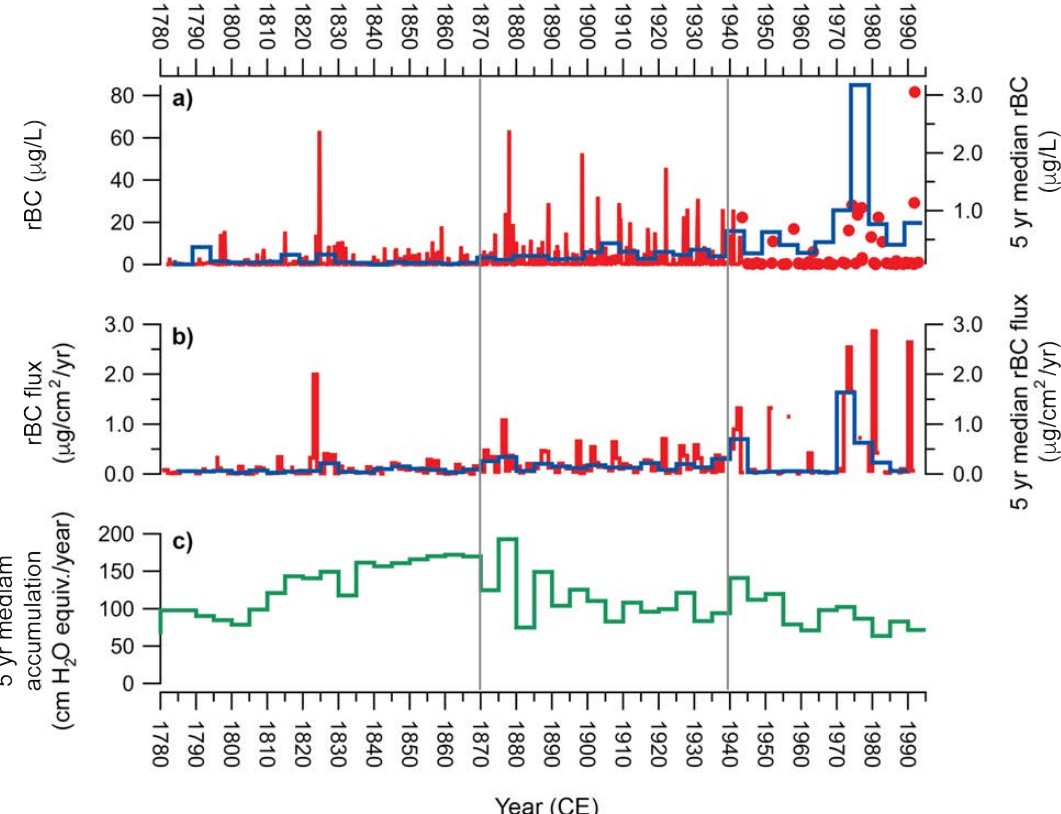


Fig. 2: a) the rBC record from the Dasuopu ice core (red). Red dots indicate discrete firn
samples. The 5 year median is indicated (blue); b) the rBC deposition flux onto
Dasuopu glacier (red) with the 5 year median (blue); c) the annual snow accumulation
record for the Dasuopu ice core (Davis et al., 2005).



Figure 3:

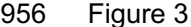

Fig. 3: Peaks in the rBC record compared to the total dust and $\delta^{18}O$ records (Thompson
et al., 2000) over 3 time intervals (a: 1819 - 1830, b: 1876 - 1890, c: 1911 - 1921 CE) in
the Dasuopu ice core. Note that peaks in the rBC record are associated with depleted
$\delta^{18}O$ and increased dust deposition.

Figure 4:

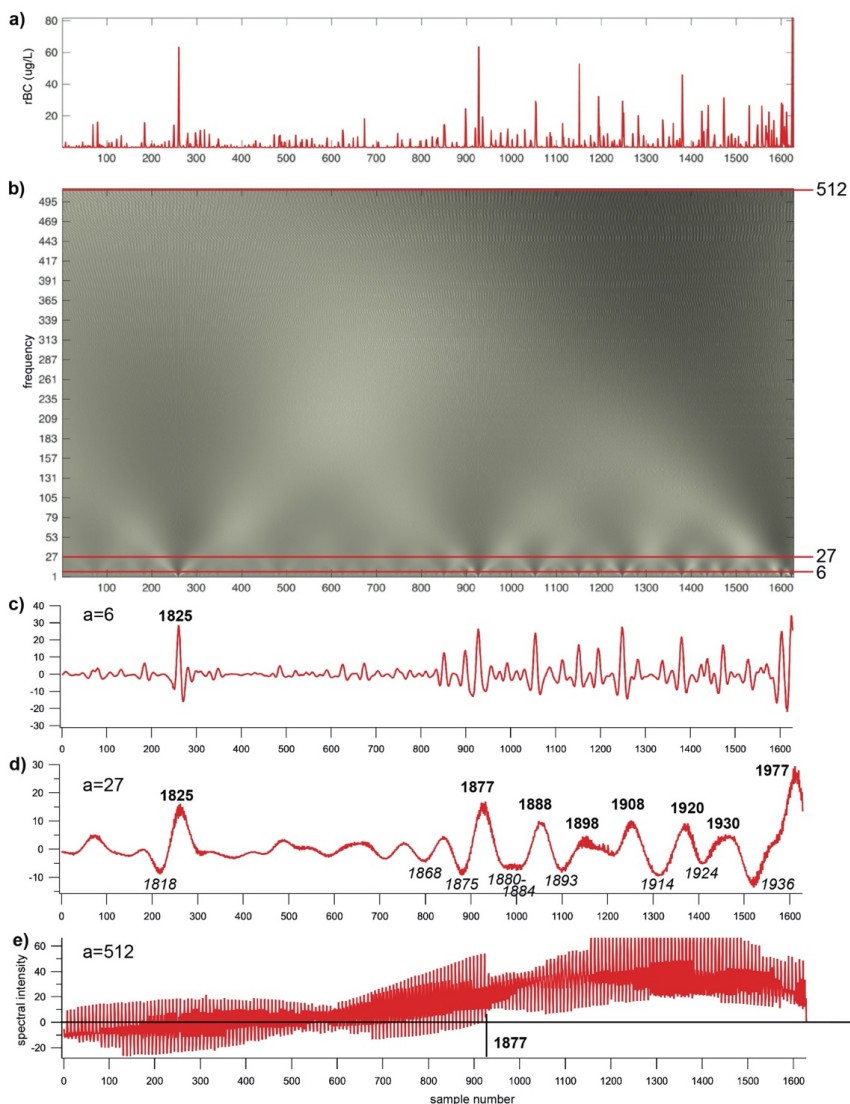


Fig. 4: The spectral analysis of the Dasuopu rBC concentration record. Sample number
1 is located at the bottom of the ice core (1781 CE) and sample number 1628 is at the
top of the firn section (1992 CE). a) is the rBC record plotted relative to sample number;
b) is the spectral analysis showing variance across all frequency scales relative to
sample number ranging from a = 2 to a = 512. Darker shades indicate relatively
stronger (more positive) coherence between the wavelet and the rBC record, as
indicated in the spectral coefficients; c, d, e) are the spectral coefficients relative to
sample number for frequency scales a = 6, 27, and 512 respectively.



Figure 5:

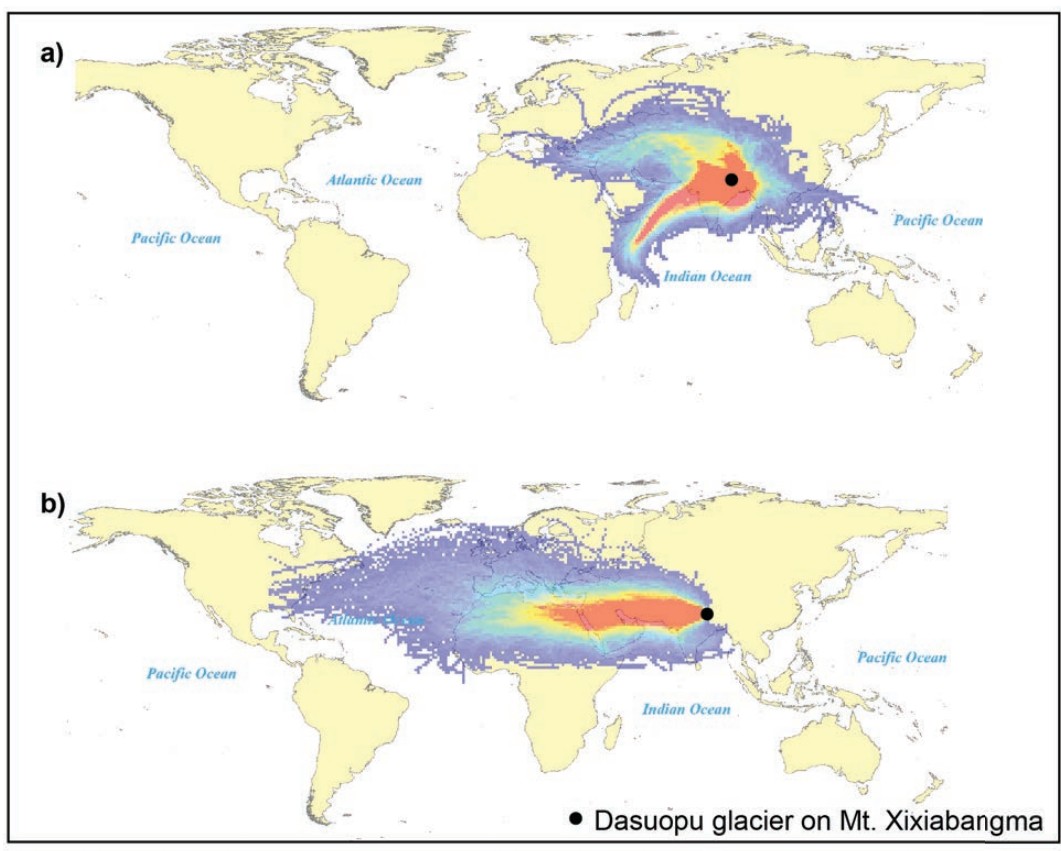

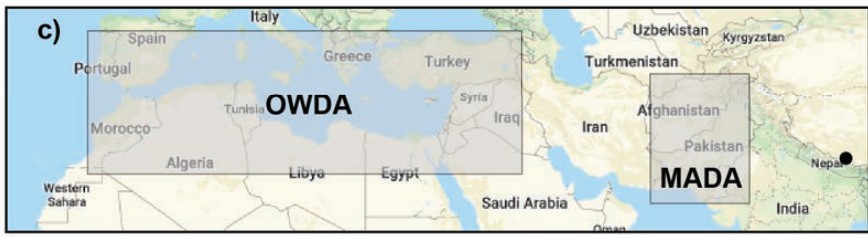


Fig. 5: Frequency of back trajectories for airmasses arriving at the Mt. Xixiabangma a)
July, b) January. The area included in the Old World Drought Atlas (OWDA; Cook et al.,
2015) and the Monsoon Asia Drought Atlas (MADA; Cook et al., 2010) reconstructions
is indicated (c).



Figure 6:

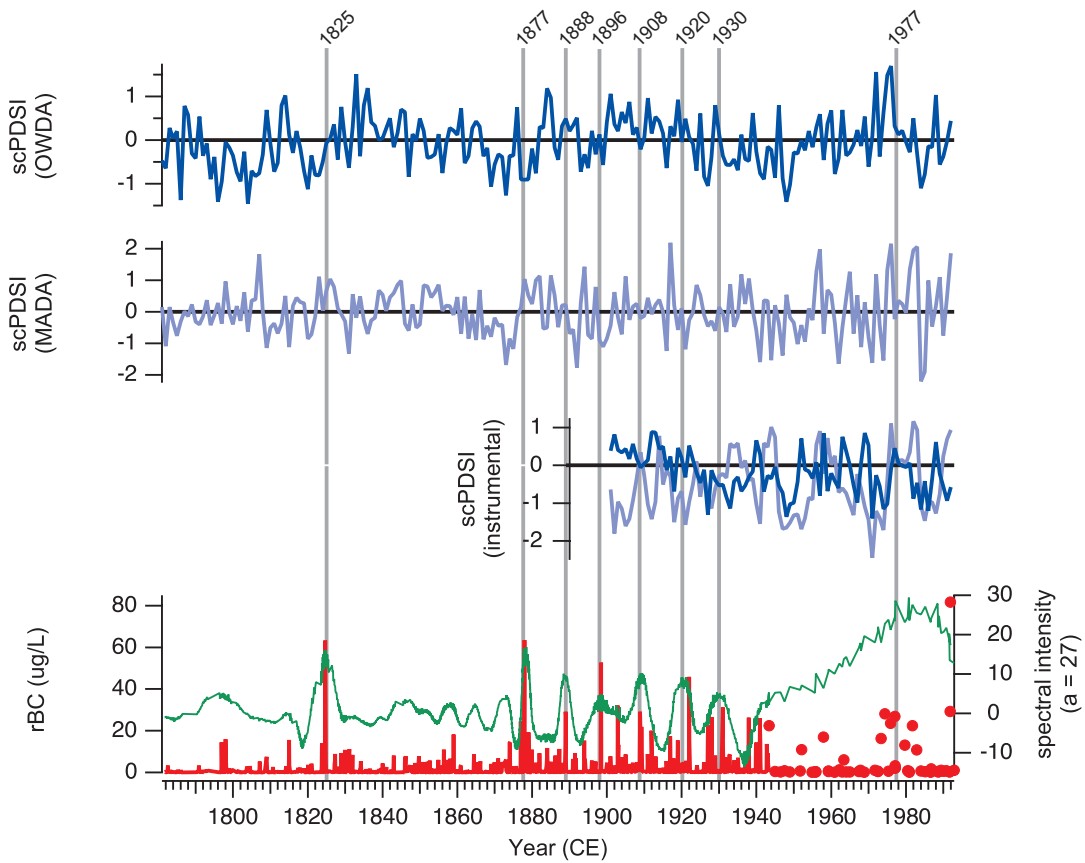


Fig. 6: The Dasuopu rBC record (in red) compared to regional reconstructed and
instrumental climate records from the Old World Drought Atlas (OWDA; dark blue) and
the Monsoon Asia Drought Atlas (MADA; light blue). Note the correspondence between
negative self-calibrating Palmer Drought Severity Index (scPDSI) and periods of high
rBC deposition. Data for both the reconstructed and instrumental climate records are
obtained from; OWDA (drought.memphis.edu/OWDA/) and MADA
(drought.memphis.edu/MADA).

Figure 7:

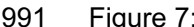


Fig. 7: The distribution of meteorologic subdivisions in NW India reporting drought
during periods of high spectral intensity at a = 27 scale.



Figure 8:

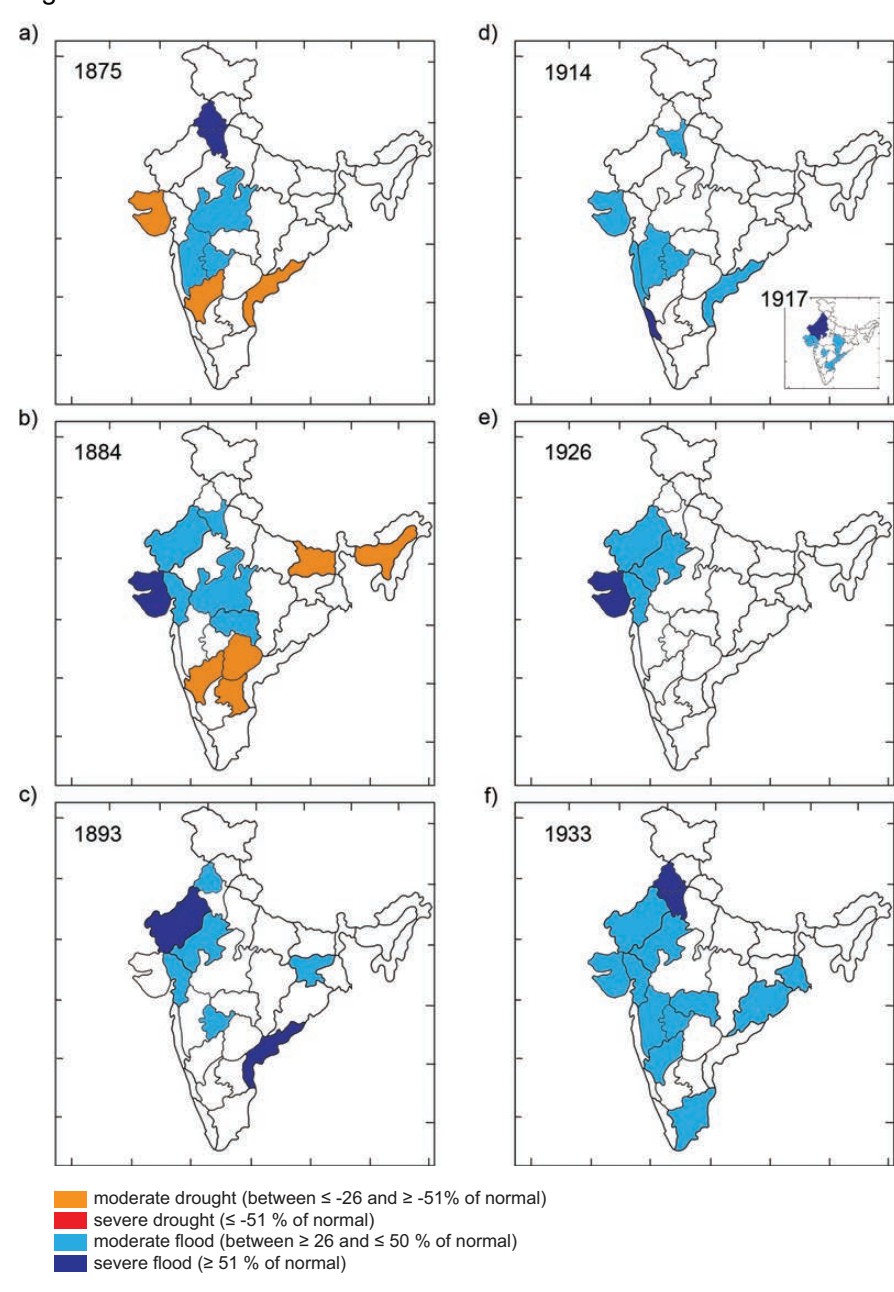


Fig. 8: The distribution of meteorologic subdivisions in NW India reporting flood
conditions during periods of low spectral intensity at a = 27 scale.





Table 1:

| trace element (n) | total $r_s$ | pre-1877 $r_s$ | post-1877 $r_s$ | EF total $r_s$ | EF pre-1877 $r_s$ | EF post-1877 $r_s$ |
|---|---|---|---|---|---|---|
| Al (915) | 0.22 | 0.40 | *-0.08* | -0.45 | -0.40 | -0.55 |
| As (914) | 0.23 | 0.41 | *-0.06* | -0.41 | -0.41 | -0.44 |
| Ba (916) | 0.26 | 0.43 | *-0.07* | -0.24 | -0.25 | -0.28 |
| Bi (857) | 0.20 | 0.40 | *-0.10* | -0.37 | -0.33 | -0.44 |
| Cd (916) | 0.23 | 0.37 | *-0.07* | -0.5 | -0.48 | -0.62 |
| Co (915) | 0.23 | 0.41 | *-0.09* | -0.38 | -0.40 | -0.42 |
| Cr (915) | 0.19 | 0.38 | *-0.09* | -0.56 | -0.53 | -0.64 |
| Cs (913) | 0.25 | 0.41 | *-0.04* | -0.39 | -0.35 | -0.48 |
| Fe (915) | 0.23 | 0.42 | *-0.07* | | | |
| Ga (915) | 0.22 | 0.39 | *-0.07* | -0.57 | -0.54 | -0.68 |
| Mg (915) | 0.24 | 0.44 | *-0.09* | -0.21 | -0.20 | -0.22 |
| Mn (915) | 0.24 | 0.44 | *-0.10* | *0.02* | *-0.01* | *0.06* |
| Mo (915) | 0.22 | 0.37 | *-0.08* | -0.54 | -0.52 | -0.63 |
| Nb (915) | 0.21 | 0.36 | *-0.04* | -0.48 | -0.46 | -0.59 |
| Ni (915) | 0.22 | 0.39 | *-0.09* | -0.5 | -0.50 | -0.57 |
| Pb (916) | 0.23 | 0.40 | *-0.08* | -0.31 | -0.31 | -0.35 |
| Rb (914) | 0.27 | 0.43 | *-0.05* | -0.49 | -0.47 | -0.60 |
| Sb (916) | 0.19 | 0.38 | *-0.07* | -0.56 | -0.52 | -0.65 |
| Ti (914) | 0.23 | 0.41 | *-0.08* | -0.28 | -0.25 | -0.42 |
| Tl (916) | 0.24 | 0.42 | *-0.08* | -0.52 | -0.49 | -0.62 |
| U (916) | 0.24 | 0.41 | *-0.07* | -0.29 | -0.29 | -0.34 |
| V (915) | 0.24 | 0.40 | *-0.07* | -0.52 | -0.51 | -0.63 |
| Zn (915) | 0.15 | 0.26 | *-0.06* | -0.53 | -0.52 | -0.63 |


Table 1. The Spearman correlation coefficient ($r_s$, $\alpha$=0.01) for trace elements and the
trace element enrichment factor (EF) relative to rBC concentration throughout the
Dasuopu ice core. Italics indicate a non-statistically significant $r_s$.