# Peer review of "Drought-induced biomass burning as a source of black carbon to the Central Himalaya since 1781 CE as reconstructed from the Dasuopu Ice Core."

_Atmospheric Chemistry and Physics, 2020_

## Referee Comment (RC1) · Anonymous Referee #1 · 3 Nov 2020

Manuscript review Title: Drought-induced biomass burning as a source of black carbon to the Central Himalaya since 1781 CE as reconstructed from the Dasuopu Ice Core Author(s): Joel D. Barker et al. MS No.: acp-2020-1052 MS Type: Research article

General comments :

The manuscript proposed by Barker and al. presents a new rBC profile reconstructed from the Dasuopu ice core covering the period 1781-1992. After an analysis of this archive using SP2 technic, a spectral analysis of the rBC profile provides an interpreta-

tion highlighting periods of drought and flooding in the emission regions. If the periods of drought are assimilated to an intensification of biomass fires emitting BC, this result is not confirmed by the contribution of trace metals.

This work upgrades this research theme with at least three significant contributions. First, a new BC record for the Himalayan region, particularly its southern barrier impacted by emissions from the Indian basin, and whose temporal coverage is large enough to include a pre-industrial period. It is however very regrettable that the post-1944 period could not be sampled with sufficient resolution and on a continuous basis for the study of this period where the anthropogenic impact is the strongest. This partially reduces the interest of the work.

The second strong point of this work corresponds to the statistical processing and spectral analysis tool applied to the BC profile. This method provides non-subjective information on the identification of events and trends that should be applied more widely in this community.

The last contribution I would like to highlight concerns the inventory and exploitation of meteorological data and other databases correlated with the results.

I suggest that this work can be accepted for publication with minor revisions and some additional information corresponding to the specific comments.

As a non-English native reviewer, I would not allow myself to make language corrections.

Specific comments :

L.43 : (Sakai and Fujita, 2017) focus on Himalayan glacier, whether precise it in the text, or add additional reference for global glacier.

L.179 : the presentation of the discontinuous sampling of this firn zone is important since it has consequences on the results presented later. It would be appreciated to know more precisely the available zones and to which season they correspond (mon-

soon, dry or intermediate season). One possibility would be to illustrate it with other parameters (water stable isotopes or dust), like done in Fig.3 for deeper sections.

L.192 : If sample were melted in their storage bag before transfer into the tubes, can you estimate the part of BC lost during this procedure as the bags were not sonicated before transfer ?

L.195 : Can you indicate the nebulization efficiency for the CETAC ?

L.201 : Can you indicate the range of concentration for your calibration ?

L.211 : "The record of rBC concentration with depth through the Dasuopu ice core provides a time series of rBC deposition onto Dasuopu glacier over time". Switching from a concentration (g.m-3) profile to a deposition (g.m-2) profile is not as trivial and requires a precise understanding of the deposition and post-deposition processes to reduce a factor of dimension. These aspects do not seem to be addressed at this stage of the study and it is therefore preferable to continue to present these results as concentrations. If I can afford it, it is possible to switch to annual deposition fluxes (g.m-2.y-1) the precise dating is available (m.y-1). "Deposition" is written elsewhere (L.229, . . .), so check them all over the manuscript.

L.284 : "the effect of discontinuously sampling the firn section and its accurate characterization of rBC since 1944 is unknown". This is true, and it has consequences for the discussion of the results obtained during this period of the record. I would suggest, if they are not simply taken out of the discussion, to use them with extreme caution, especially if they are confronted with other continuous sampling periods or similar periods from other sites and glaciers. Figure 2b illustrates the annual flux of rBC. I imagine, for lack of a description, that it is calculated and illustrated only for complete years, without sampling discontinuity. Moreover, for the concentrations and fluxes in Fig. 2a & 2b, since annual values for the period after 1944 are missing, the median value over 5 years is not representative (despite your explanations).

L.297-300 : see previous comment.

L.303 : Not being a specialist of this spectral analysis, you can bring more precisions on the method of identification of the 3 modes at 6, 27 and 512 beyond the simple visual aspect of figure 4b.

L.304-309 : What is the robustness of this analysis on the discontinuous section after 1944 ? Can it be made consistent with the previous zone ? Some border effect as suggested at L.498 ?

L.316 : From Fig. 4e only, I can't identify this shift between positive and negative coefficients at 1877 without further explanations ... Once this point has been clarified, I appreciate this approach using spectral analysis.

L.365 : As done in Ginot et al (2014) using Lim et al. (2014) information, the impact of melting/freezing rBC loss could be approached to compare with Kaspari data. In any case, it is difficult to compare sites on the basis of the average concentration value obtained since several processes are involved (sources, transport, accumulation, deposition, post deposition ...). On the other hand, the comparison of trends, increasing factor during anthropisation or particular events remains appropriate.

L.375 : This analysis on the seasonality of the rBC does not bring much new information compared to the referenced results. By using statistical and spectral analysis tools ... that you seem to master, and by compiling the different years, I am convinced that you can deepen this point. A fine analysis of seasonality would strengthen the conclusions on the identification and impact of biomass fires concentrated in one season, while other sources of BC are more temporally distributed over the year.

L.402-415 : BC concentration and snow accumulation are directly linked. Use only annual BC fluxes and snow accumulation for this temporal variation study and their link with climate, emissions ...

L.530-557 : Concerning correlation between rBC and trace metals, as you point out,

you do not have the right tracers associated with biomass fire to provide meaningful interpretation. However, some inorganic tracers (potassium?) should have been analyzed by ion chromatography at BPRC.
* * *

---

## Referee Comment (RC2) · Anonymous Referee #2 · 16 Nov 2020

This manuscript presents an ice-core-based black carbon (BC) record starting in 1781 CE from Dasuopu glacier in central Himalaya. BC is deposited is mainly deposited during the non-monsoon season. An increase in BC concentration and frequency after 1877 was observed and attributed to biomass burning associated with periods of drought in neighbouring regions. BC records are scarce from that region and are needed to constrain emission estimates used in modelling the aerosol effect on climate. Generally, the manuscripts is well written and structured and mostly scientifically sound (see comments below). The manuscript deserves publication, after taking into

account the comments and suggestions listed in the following.

Presentation of the data and spectral analysis: To give the reader the chance to assess the quality of the data, the rBC concentration should be shown first on a depth scale before transformation onto a time scale. To me it is not clear what is given in fig. 2a and all the subsequent figures. Are those raw data or annual averages? Considering the variable sampling resolution with depth and time (as shown in fig. S1), only averaged data (e.g. annual averages) can be interpreted with respect to signal intensity and frequency. This also applies to the spectral analysis. The principle of spectral analysis is that values represent similar time intervals. Please clarify and correct if necessary.

Attribution of BC to biomass burning: I was surprised to see that BC was purely interpreted as emitted from biomass burning. Potential contributions from combustion of fossil fuels are not discussed. This is in contrast to the emphasis given to anthropogenic BC in the introduction, and to the interpretation of other ice core-based BC records from the Himalaya (Ming et al., 2008, Kaspari et al., 2011). The latter show rather similar trends with peak values in 1970s and 1990s and a minimum in the late 1980s, which is also visible in the Dasuopa record. I suggest including a comparison with those records in the manuscript, since they are located so closely. The biomass burning hypothesis is also in contradiction to the interpretation of trace metal enrichment being related to fly ash from European Industrial Evolution (Gabrielli et al., 2020, partly the same authors). If larger fly ash particles were transported that far, why not BC? There are BC records from Europe to compare with (Lim et al. 2017, Sigl et al. 2018). The attribution to biomass burning is based on the comparison with the Palmer Drought Severity Index. This is an interesting idea. As indication, individual events are discussed, when they match or do not match, which is a rather descriptive section and hard to follow. To convince the reader about this hypothesis try to use a more robust statistical approach, for instance a spatial correlation analysis with the PDSI maps. In summary, the biomass burning hypothesis needs a more convincing argumentation.

The effect of BC on glaciers is discussed in the introduction and was presumably a

motivation for this study. Overall, the BC concentrations are rather low. What does this mean for the albedo? This deserves a statement in the conclusions.

BC analysis: Losses of BC in melted snow/ice have been observed before. I am a bit worried about your transfer of the melted sample from the storage bags into the 50 ml polypropylene centrifuge tubes. Did you check if you have losses during this step? Explain Aquadag standards. Define baseline conditions and background level.

BC shows often a pronounced seasonality (see for example Ginot et al. 2014) as you show as well for the Dasuopu record in fig. S2. It would be interesting to see in a figure if this is the case throughout the continuous record. Have you cross-checked the previous dating with annual layer counting using the BC seasonality and if not why not?

The introduction is overall very descriptive. Could you add some summarizing?

Minor comments

L 57: "the European Industrial Revolution", why only European?

L 99: "Similarly, Liu et al. (2008) report high elemental carbon (a form of BC)". EC is not a form of BC, be more precise about the difference.

L 118: What is the importance of being the highest, explain.

L 127: What do you mean with mixed free tropospheric composition? Explain. What is the percentage of real free tropospheric conditions over the year?

L 136: I suggest to include the earlier paper on BC terminology: Petzold, A., et al., Recommendations for reporting "black carbon" measurements, Atmos. Chem. Phys., 13, 8365–8379, https://doi.org/10.5194/acp-13-8365-2013, 2013.

L 137: "sub-annual resolution in the glacier ice portion". What do you mean by that?

L 152: "Here, we examine the upper section of the C3 ice core". Give more details on how many cores were collected. Was the chronology developed for the same core? If

not, comment on how the cross-dating was conducted.

L 174: Here and in entire manuscript: Rephrase: "ultra-pure water", and add quality.

L 184: Add concentration of acid.

L 186: Ziploc bags. Ziploc is a brand, replace by material of the bag.

L 249-258: Is this data from Gabrielli et al. (2020)? If yes, this has to be stated.

L 261: "topography" instead of "geomorphology"

L286: For comparison add the percentage of snow deposited during monsoon conditions in the ice section.

L 289: rBC concentration and not deposition.

L 289-291: This is a rather qualitative statement made only for selected times periods. Can you support that with correlation coefficients over the entire record?

L 551: all of the trace element EFs

Fig. 2: The 5-year median looks strange to me, e.g. at 1790 and 1975 and generally in the firn part the median is higher than any single value. Please check.

Fig. 5: Add colour code values for the frequencies.

References Ginot, P., et al. (2014). "A 10 year record of black carbon and dust from a Mera Peak ice core (Nepal): variability and potential impact on melting of Himalayan glaciers." The Cryosphere 8(4): 1479-1496.

Lim, S., et al. (2017). "Black carbon variability since preindustrial times in the eastern part of Europe reconstructed from Mt. Elbrus, Caucasus, ice cores." Atmos. Chem. Phys. 17(5): 3489-3505.

Sigl, M., et al. (2018). "19th century glacier retreat in the Alps preceded the emergence of industrial black carbon deposition on high-alpine glaciers." The Cryosphere 12(10):

3311-3331.

---

## Author Comment (AC1) · 15 Feb 2021

We thank Referee #1 for their thoughtful and insightful comments. We have responded to their comments in the manuscript as described below.

L 43: Thank you for alerting us to the fact that the cited reference focusses on Himalayan glaciers only. As our focus is on glaciers on a global scale, we have removed the Sakai and Fujita (2017) reference and instead cited Gregory and Oerlemans (1988) for the global response by glaciers to warming, Xu et al. (2012) for

albedo effects on glacier melt, and Raper and Braithwaite (2006) for a decrease in precipitation as snow as being a factor for glacier mass loss. Further, we have modified the text to state that "While warming summer temperatures resulting in increased glacier mass loss (e.g. Gregory and Oerlemans, 1998) and decreasing precipitation as snow (Raper and Braithwaite, 2006) are important factors contributing to glacier mass wastage globally. . .".

L179: We welcome Referree #1's recommendation and have included the recommended figure as Suppl. Fig. 4.

L192: This is a point that Referee #2 raised as well. No, we are unable to estimate the portion of BC that could potentially be lost during sample melting in a polyethylene sample bag. However, Wendl et al. (2014) report that no significant BC particle loss occurs until ∼3 days of storage in polypropylene vials at room temperature, and that there is less adherence at cooler temperatures. Lim et al. (2014) confirm these results and indicate that melting at room temperature is preferable to melting in a warm bath. Meinander et al. (2020) suggest that some EC adherence to polyethylene sample bags may occur, but that EC heterogeneity in the sample (in this case snow) exceeds any particle adherence that may have occurred.

Our ice samples were melted over the course of no more than an hour (less than 3 days) and our samples did not reach room temperature prior to transfer to the polypropylene vial followed by sonication. While we cannot rule out the possibility of particle adherence to the polyethylene sample bags during melting, we suggest that any adherence would be minimal because of the short melting period (< 1 hour) at cool temperatures (< room temperature). Further, we suggest that the extent of any possible particle adherence would be similar between individual samples because all samples were melted in an identical manner.

L 198: We have added the nebulization efficiency for the nebulizer (U5000) based on 2 references that are provided as follows: ". . .CETAC U-5000AT+ ultrasonic nebulizer

(Teledyne CETAC Technologies, Omaha, U.S.A.; ~18% nebulization efficiency at 220 nm, 356 nm, and 505 nm particle size (Menking, 2013, Wendl et al., 2014))..."

L205: Thank you for bringing this to our attention. We have added test stating the calibration standard range as follows "A 5-point calibration curve (~0.75 – 12.5 ppb)...".

L233: This is a point that Referee #2 highlighted as well. Thank you for bringing this to our attention. We did not intend to use "deposition" as a quantitative entity here, but rather as a process by which rBC in the atmosphere is transferred onto the glacier surface for incorporation into the firn and glacier ice. For clarity, we have rewritten the text as "...characterization of the rBC deposited onto the Dasuopu glacier...". We have also added clarification where the term "deposition" occurs elsewhere in the manuscript, using the word "concentration" when referring to rBC content in the ice core, rather than "deposition".

L211: We are unsure what Referee #1 is referring to here. The flux as plotted in Fig 2b is an annual flux (ug/cm2/yr).

L284: The point that we should use post-1944 (discontinuous firn sampling) data with extreme caution is well taken. With this in mind, we note that the core has "elevated [rBC] concentrations during the 1960s and 1970s" (line 302-303), and while we cannot comment with certainty regarding the frequency of these high rBC events, the observation that high rBC concentrations are present in the firn section is supported by the data presented. We are careful to caution the reader that the discontinuous sampling in the firn section presents problems in comparing rBC concentrations and fluxes to other ice core records over this time period (line 561-562).

L284: Thank you for pointing out our error in including the discontinuous firn data in the median calculations for both the rBC record and the rBC flux. Fig 2 has been modified to include only the continuous ice section in the Dasuopu core.

L303: Thank you for highlighting the need for clarification here. At scale a = 6, we

are focusing on the ∼annual scale in the rBC record. At a = 27 (∼4.5 years) we are focusing on sea surface temperature (SST) oscillation in the Cape Hatteras region of the North Atlantic which should influence westerly circulation to the Dasuopu cite during winter months. 4.5 years has been identified as the middle value of 3 modes of SST oscillation by Feliks et al. (2011). At a =512, we detect a large scale shift using a scale encompasses the entire record in one pass. Text to clarify our choice of modes has been added to Section 3.2 as follows: "We chose to examine 3 modes of variability within the spectral analysis (Fig. 4b), 2 of which correspond to North Atlantic sea surface temperature (SST) because of the important role of westerly atmospheric circulation in the Dasuopu region during the winter non-monsoonal season (Davis et al., 2005); the annual frequency that is that is responsible for 90% of the variance in the seasonal cycle of SST in the North Atlantic (Feliks et al., 2011), and ∼4.5 year variability that is the middle value of 3 modes of SST oscillation (3.7, 4.5, and 6.2 years; Feliks et al., 2011) in the Cape Hatteras region of the North Atlantic (44 °N, 47 °W). A third mode of variability (∼85 years) was chosen to identify longer-term variation in the rBC record."

L304-309: This is a good point and we deliberately constructed the spectral analysis to take discontinuities in the ice core into account. We constructed the data input for the spectral analysis to consider each rBC measurement as an individual data point rather than being tied to a linear time distribution. As such, while the rBC measurements are arranged chronologically, they do not represent a specific point in time. The assignment of "time" occurs as the last step in the process when we examine which samples are associated with phenomena identified by the spectral analysis. As such, data gaps or discontinuous sampling minimally influences the spectral output, and the discontinuous firn section can be included in the analysis. We are aware of possible border or "edge" effects. Edge effects will be most noticeable at larger scales where the wavelet is truncated at the edges of the dataset. Conversely, edge effects are less at finer scales were the wavelet itself is "smaller" and thus less truncated at the edge of the dataset. Consequently, we do not make strong inferences from the results

of the spectral analysis near the end-points of the dataset, particularly at large scale (L509-516).

L316: Fig. 4e shows that the spectral coefficients do not dip below 0 after ∼1877 CE (ie: the shift to there being no negative coefficients occurs at ∼1877 CE). This point is highlighted in the figure. We have added the text "The a = 512 (∼85 year) mode identifies a shift from some samples with negative spectral coefficients (values below zero) to those with positive spectral coefficients at ∼1877 CE (Fig. 4e)." (L330) for clarification.

L375: We appreciate this point and this is an issue that we struggled with when writing the manuscript. It is difficult to demonstrate a process occurring at an annual or seasonal scale over the broad period covered by the ice core. This is why we chose 3 intervals to highlight the relationship between isotopic composition, dust concentration, and rBC concentration. In an effort to show this relationship over a broader analytical "window" we have included a new analysis using spectral coherence of rBC concentration and d18O over a ∼50m section of the ice section. This analysis shows the strength of correlation between d18O and rBC concentration at multiple period scales as well as any phase lag in this correlation. We hope that this proves to be a more effective way of showing the seasonality of rBC concentration though this section of the ice core.

L402: We are unsure what Referee #1 is referring to here. Fig 2b shows the rBC flux as an annual flux.

L530-557: We appreciate Referee #1's suggestion here. Originally, only anions were analyzed by Thompson for the Dasuopu core, and although we were able to obtain unpublished potassium data for the Dasuopu core, we were cautioned that its accuracy was questionable. Analysis of this suspect record indicated that there is no correlation between K+ and rBC at raw, 1 standard deviation from the mean and 2 standard deviations from the mean, and given that the result does not alter the conclusions, and that the K record is of questionable quality, we did not include it in the submitted manuscript.

---

## Author Comment (AC2) · 15 Feb 2021

We thank Referee #1 for their thoughtful and insightful comments. We have responded to their comments in the manuscript as described below.

We appreciate Referee #2's point regarding presenting data vs depth before presenting data vs time. However, the rBC data, for example Figure 2a, is presented using the Thompson et al (2000) time-depth chronology that was established using d18O, dust, and NO3- measurements and annual layer counting confirmation using the location of

the 1963 CE beta radioactivity peak, and further calibrated using 2 major monsoon failures at 1790-1796 and 1876-1877 as benchmarks (L156-164). This chronology is used in Thompson et al (2000). Data in this publication are presented relative to time (years AD), and not depth. We have adopted the same convention.

Fig 2a presents raw data and 5 year median data. We do not specify that these are annual averages.

The point that "only averaged data can be interpreted with respect to signal intensity" is well taken and something that we considered during the data analysis. However, we purposefully performed the spectral analysis using individual samples rather than annual averages for precisely the reason that the Referee mentions. By inputting data as dimensionless samples (with respect to time, depth, annual layer thickness, sampling resolution) in chronologic order, we are a) not introducing artifacts due to variable sampling resolution and annual layer thickness; b) we preserve the signal of sudden increases and decreases in rBC concentration that is inherent in the dataset and is an important feature of the rBC record (this information is greatly muted using annual averages). We agree that the principle of spectral analysis is that values represent similar intervals, and we suggest that these intervals need not be "time" as suggested by Referee #2, but instead that the intervals can be similar entities (for example samples, or rBC concentrations), as we've done here.

Referee #2 states that they are "surprised that BC was purely interpreted as emitted from biomass burning. Potential contributions from combustion of fossil fuels were not discussed". We thank the Reviewer for their perspective, but we believe that we do not discount a potential contribution to the rBC record from fossil fuels, but rather show that a significant contribution from this source is not supported by available trace element data and rather aligns strongly with records or regional drought and, by extension, biomass burning. Most other studies examining BC records from ice cores in the region find that contributions from fossil fuels increase in the 1970s, which is a time period that is not well resolved in the Dasuopu BC record. We simply point out that the

trace element and drought indices suggest that biomass burning may be an important source of rBC. We agree that our presentation is descriptive and we value Referee 2's suggeswion that we use a "more robust statistical approach" to support the correlation between regional drought and periods of high rBC concentration in the Dasuopu core. Unfortunately, in the time provided, we have been unable to conduct an analysis such as a spatial correlation analysis between the regions described in the PDSI maps and the Dasuopu glacier drilling cite. However, we would like to point out that we provide evidence from a trace element record, as well as 3 independent climate records, so support our conclusions that rBC may be associated with dry conditions and associated biomass burning events.

We agree with Referee #2 that a direct comparison between the rBC record presented here for the Dasuopu ice core and the cores presented for East Rongbuk glacier by Ming We agree with Referee #2 that a direct comparison between the rBC record presented here for the Dasuopu ice core and the cores presented for East Rongbuk glacier by Ming et al. (2008) and Kaspari et al. (2011) is an excellent idea, we are limited by the discontinuous sampling of the firn layer of the Dasuopu core that spans the time periods presented by the East Rongbuk cores. We do provide a qualitative comparison on lines 303-305 where we state that "The discontinuous firn section of the core has elevated concentrations during the late 1960s – 1970s, consistent with observations from East Rongbuk glacier by Ming et al. (2008) and Kaspari et al. (2011) , and for Tanggula glacier by Xu et al. (2001)." This comparison, as well as a comparison with other glacier cites is shown in Fig. 1.

Referee #1 also questioned whether we had assessed BC particle loss during sample melting. I have repeated our response here: "No, we are unable to estimate the portion of BC that could potentially be lost during sample melting in a polyethylene sample bag. However, Wendl et al. (2014) report that no significant BC particle loss occurs until $\sim$3 days of storage in polypropylene vials at room temperature, and that there is less adherence at cooler temperatures. Lim et al. (2014) confirm these results

and indicate that melting at room temperature is preferable to melting in a warm bath. Meinander et al. (2020) suggest that some EC adherence to polyethylene sample bags may occur, but that EC heterogeneity in the sample (in this case snow) exceeds any particle adherence that may have occurred.

Our ice samples were melted over the course of no more than an hour (less than 3 days) and our samples did not reach room temperature prior to transfer to the polypropylene vial followed by sonication. While we cannot rule out the possibility of particle adherence to the polyethylene sample bags during melting, we suggest that any adherence would be minimal because of the short melting period (< 1 hour) at cool temperatures (< room temperature). Further, we suggest that the extent of any possible particle adherence would be similar between individual samples because all samples were melted in an identical manner."

We appreciate the suggestion that we cross check the annual layer counting with seasonal rBC increases in the Dasuopu rBC record. This is something that we attempted during our data analysis. In the end, we found that the presence of missing rBC samples (as described in the manuscript) introduced error to the cross-dating effort, rendering it unreliable.

L 57: Thank you. We have changed "European Industrial Revolution" to "Industrial Revolution".

L 99: Thank you, we have removed the false statement that EC is a form of BC.

L118: We explain that the importance of the Dasuopu core being from high elevation is that it allows us to sample from the free troposphere, distant from local sources of BC contamination (lines 126-134).

L 127: Implicit in the term "free troposphere" is that we are not influenced by sources of local BC contamination. The drill site on Dasuopu glacier is not influenced by down-valley meteorological conditions, as described by Li et al (2011) and cited in

the manuscript). While not measured for this location specifically, and so not noted in the manuscript explicitly, the free troposphere in the central Himalaya begins at ~2.5 km elevation in the winter and 3.3 km elevation in the summer (Solanki and Singh, 2014). We have added this sentence for clarity (Line 135): "Generally, the lower limit of the free troposphere in the central Himalaya occurs at ~2.5 km in the winter and 3.3 km in the summer seasons (Solanki and Singh, 2014)."

L 136: Thank you. We have added this reference to line 138.

L 137: By "sub-annual resolution" we mean that we were able to obtain multiple samples per year of accumulation. We have replaced "sub-annual" with "seasonal" for clarity.

L 152: Information regarding the recovery of Dasuopu core 3 is described in detail in the Thompson et al (2000) publication, that is referenced in the manuscript. Thus, we believe that repeating this information is unnecessary and beyond the scope of this manuscript. Likewise, as described my Thompson et al (2000), the chronology was developed for core 3 (used here).

L 174: We're not sure that we ever used the term "ultra-pure water". We do write MQ water, and have replaced this initial description with "type 1 Milli-Q water" for clarity.

L 184: Thank you. We have added "2N" descriptor to HNO3.

L 186: Thank you. We have replaced the brand "Ziploc" with "polypropylene".

L 249-258: Yes, this is the data that Gabrielli et al (2020) used, and we have clarified this by writing "using methods described in Uglietti et al. (2014) and reported by Gabrielli et al. (2020)."

L 261: Thank you. We have replaced "geomorphology" with "topography".

L 286: We think that this is a good suggestion, but at this point, we think that the effort involved to determine how much snow was deposited during the monsoon for the 64 m

ice section of the core analyzed here for a piece of information that doesn't contribute strongly to the findings of the manuscript is unwarranted.

L 289: Our mis-use of the term "deposition" throughout the manuscript was noted by Referee #1 as well. We have replaced "deposition" with "concentration" throughout the manuscript.

L 289-291: The point that we approach the relationship between d18O content, dust, and rBC in a qualitative way for discrete intervals in the core was noted by Referee #1 too. We respond to their, and your observation as follows: "manuscript. It is difficult to demonstrate a process occurring at an annual or seasonal scale over the broad period covered by the ice core. This is why we chose 3 intervals to highlight the relationship between isotopic composition, dust concentration, and rBC concentration. In an effort to show this relationship over a broader analytical "window" we have included a new analysis using spectral coherence of rBC concentration and d18O over a ~50m section of the ice section. This analysis shows the strength of correlation between d18O and rBC concentration at multiple period scales as well as any phase lag in this correlation. We hope that this proves to be a more effective way of showing the seasonality of rBC concentration though this section of the ice core."

L 551: Thank you for noting that omission. We have corrected the sentence as per your suggestion.

Fig 2: The secondary x axis (the 5 year median scale) is different than the rBC scale on the primary axis. I believe that this is the source of the confusion.

Fig 5: Thank you for pointing this out. We have added the following to the Figure caption: "Red and blue indicate a higher frequency and lower frequency air mass flow paths, respectively".

---

## Author Response (AR2)

Dear Sir of Madame- Thank you very much for bringing these two issues to our attention. We have addressed them directly as described below.

-
-  In an effort to more directly address why we cannot resolve increases in rBC concentrations during the 1970s as observed in Himalayan ice cores and attributed to increases in regional fossil fuel emissions, we have added text as follows:
    o Line 587: "Of importance is that the discontinuous sampling of firn in the Dasuopu ice core record presented here does not capture a continuous record of rBC deposition during the post 1970s; a period when rBC is reported to have increased in the southern Himalaya (Kaspari et al., 2011) and Tibetan Plateau (Jenkins et al., 2016; Wang et al., 2015) in response to regional increases in fossil fuel emissions. An evaluation of the contribution of fossil fuel emissions to the rBC record in the Dasuopu ice core, particularly during the 1970s, is compromised here by the insufficient sampling resolution in the firn section of the ice core."
    o Line 615: "It should be noted that the Dasuopu ice core rBC record is discontinuous during the period of increased regional industrial activity *(during the 1970s)* thus the available data cannot address the importance of this regional industrialization to rBC deposition onto Dasuopu glacier."

- To address Referee #2s comment regarding the implications of rBC on Dasuopu glacier albedo, we have added the following test beginning on line 610:
    o "While the focus of this study was utilizing the ice core record to investigate the role of biomass burning and drought on BC emissions, the Dasuopu BC ice core record provides information about BC induced albedo reductions at 7200 m in the high Himalaya. Generally, BC concentrations are low (mean = 1.52; median= 0.17 µg/L which would have a very modest albedo effect. However, the episodes of high BC deposition (between ~15 and 82 µg/L, as identified by the spectral coefficients associated with drought events (Fig. 6) could have a larger albedo lowering effect, with interesting implications related to drought conditions and snow melt feedbacks. Investigating albedo reductions from the Dasuopu ice core record is beyond the scope of this study, but warrants further investigation."